# A Strategy for Variable-Scale InSAR Deformation Monitoring in a Wide Area: A Case Study in the Turpan–Hami Basin, China

**Yuedong Wang** [1,2]**, Guangcai Feng** [1,2,*]**, Zhiwei Li** [1,2]**, Shuran Luo** [1]**, Haiyan Wang** [3]**, Zhiqiang Xiong** [1]**, Jianjun Zhu** [1,2] **and Jun Hu** [1,2]

1   School of Geosciences and Info-Physics, Central South University, Changsha 410083, China
2   Key Laboratory of Metallogenic Prediction of Nonferrous Metals and Geological Environment Monitoring Ministry of Education, Changsha 410083, China
3   Chongqing 208 Geological Environment Research Institute Co., Ltd., Chongqing 400700, China
*   Correspondence: fredgps@csu.edu.cn; Tel.: +86-182-7486-7449

**Abstract:** In recent years, increasing available synthetic aperture radar (SAR) satellite data and gradually developing interferometric SAR (InSAR) technology have provided the possibility for wide-scale ground-deformation monitoring using InSAR. Traditionally, the InSAR data are processed by the existing time-series InSAR (TS–InSAR) technology, which has inefficient calculation and redundant results. In this study, we propose a wide-area InSAR variable-scale deformation detection strategy (hereafter referred to as the *WAVS–InSAR strategy*). The strategy combines stacking technology for fast ground-deformation rate calculation and advanced TS–InSAR technology for obtaining fine deformation time series. It adopts an adaptive recognition algorithm to identify the spatial distribution and area of deformation regions (regions of interest, ROI) in the wide study area and uses a novel wide-area deformation product organization structure to generate variable-scale deformation products. The Turpan–Hami basin in western China is selected as the wide study area (277,000 km$^2$) to verify the proposed WAVS–InSAR strategy. The results are as follows: (1) There are 32 deformation regions with an area of $\geq 1$ km$^2$ and a deformation magnitude of greater than $\pm 2$ cm/year in the Turpan–Hami basin. The deformation area accounts for 2.4‰ of the total monitoring area. (2) A large area of ground subsidence has occurred in the farmland areas of the ROI, which is caused by groundwater overexploitation. The popularization and application of facility agriculture in the ROI have increased the demand for irrigation water. Due to the influence of the tectonic fault, the water supply of the ROI is mainly dependent on groundwater. Huge water demand has led to a continuous net deficit in aquifers, leading to land subsidence. The WAVS–InSAR strategy will be helpful for InSAR deformation monitoring at a national/regional scale and promoting the engineering application of InSAR technology.

**Keywords:** wide-area deformation; deformation detection; time-series InSAR; stacking; Turpan–Hami basin

## 1. Introduction

Advanced microwave remote sensing technology can precisely monitor deformation over wide areas, which helps geohazard surveys of phenomena such as underground fluid development, mineral mining, and landslide. In recent years, fast-developing interferometric synthetic aperture radar (InSAR) technology and abundant available synthetic aperture radar (SAR) data [1–4] has laid the foundation for high-precision and wide-scale InSAR ground-deformation monitoring. InSAR technology has been successfully used to monitor ground deformation at a regional [5–9] and national scale [10–13]. Large-scale geodetic technology, such as InSAR, usually describes the spatial characteristics of ground deformation by deformation rate, and shows deformation development over time using a time series of deformation. The deformation region usually accounts for a small part of the monitoring area [11], so the ground deformation we are interested in only accounts for a small part of

the wide-area monitoring results. At present, the engineering projects to obtain ground deformation in a wide study area (WSA) usually calculate the deformation time series using InSAR datasets covering the whole WSA, using time-series InSAR (TS–InSAR) technology. Even with multiple spatial resolutions, such schemes require a lot of computing resources and storage space, and even then require repeated calculations and provide redundant results, especially in the non-deformation region [14]. Therefore, it is necessary to develop a set of efficient monitoring methods and procedures for wide-area InSAR deformation and a more feasible multi-scale deformation product organization structure in the WSA.

One way to improve the computational efficiency of the TS–InSAR method is to introduce a parallel processing method, which can be realized using high-performance computers (HPC) [12,15–19]. However, the high cost of HPC equipment hinders the popularization of this strategy. Another way is to improve the TS–InSAR method itself, by introducing sequential adjustment theory for real-time InSAR data processing [20–22], introducing a geological model or time-series filtering algorithm for high-dimensional deformation calculation [23–26], or realizing a high-precision TS–InSAR deformation calculation using block solutions [27,28]. These strategies can improve the efficiency of the TS–InSAR solution to a certain extent. However, for wide-area InSAR deformation monitoring, high-precision independent calculation of all InSAR datasets in the WSA will provide many useless time-series results, especially in the non-deformation area. Therefore, it is necessary to develop a demand-oriented multiple spatio-temporal-scale deformation monitoring method, considering the universality of monitoring strategies, computing resources, measurement accuracy, and the efficiency of deformation calculation and interpretation.

The averaging of multiple interferograms (stacking) method was proposed by the authors in [29], which can obtain the ground-deformation rate by averaging the phase of the multitemporal differential InSAR (DInSAR) dataset. Compared with conventional TS–InSAR technologies, such as persistent scatterer (PS) [30], small-baseline subset (SBAS) [31], and interferometric point target analysis (IPTA) [32], stacking only obtains the deformation rate with lower technical requirements and higher computational efficiency. Stacking has been widely used for deformation identification [33–37]. A wide-area deformation monitoring project usually identifies deformation regions based on the ground-deformation rate [38]. For the deformation region, the corresponding deformation time series is extracted to analyze the spatio-temporal evolution of deformation. The deformation time series in stable zones has less information. Therefore, combining stacking and TS–InSAR may contribute to efficient variable-scale deformation monitoring.

In this study, we propose a wide-area InSAR variable-scale deformation detection strategy (WAVS–InSAR). WAVS–InSAR uses stacking technology to quickly calculate the low-spatial-resolution ground-deformation rate over the WSA. Then, an adaptive intelligent recognition algorithm is used to identify the location and area of the deformation regions and determine the regions of interest (ROI). Advanced TS–InSAR technologies are then used to obtain the high-spatio-temporal-resolution deformation time series in the ROI. Finally, the variable-scale InSAR deformation product in the WSA is obtained by a novel variable-scale deformation product organization structure. To verify the proposed WAVS–InSAR strategy, we applied it to the Turpan–Hami basin (about 277,000 km$^2$) in Xinjiang, China. The Turpan–Hami basin is the driest place in China, and has the least rainfall in China. Many tectonic faults, as well as agricultural and mining areas, are scattered across the basin. It is of great significance to obtain the spatio-temporal distribution characteristics of ground subsidence and to investigate the surface deformation related to the active agricultural economy and mineral exploitation in the basin.

The remainder of the paper is organized as follows. We introduce the WAVS–InSAR strategy in Section 2. In Section 3, the general situation of the Turpan–Hami basin, InSAR data, and the data-processing details are briefly described. The variable-scale deformation product in the Turpan–Hami basin is shown in Section 4, followed by the discussion in Section 5. Section 6 presents the conclusions.

## 2. Methodology

We first collect all available InSAR datasets covering the WSA, and preprocess all datasets through registration and DInSAR, to generate the multitemporal DInSAR datasets with the same spatial reference data. Then, we apply the WAVS–InSAR strategy to process the multitemporal DInSAR data to obtain variable-scale deformation products in a wide area. The WAVS–InSAR includes the following four modules (Figure 1).

(1) We obtain the wide-area deformation rate using the stacking method [29]. First, we calculate the deformation rate of each frame using stacking. Then, we mosaic the results of all frames to obtain the wide-area deformation rates.

(2) We detect ROI from the deformation rates. Setting the threshold for the deformation rate, the extension radius, and the minimum clustering area, we calculate the spatial distribution and area of the ROI in the WSA using an adaptive deformation detection method [39].

(3) We obtain the high-spatio-temporal-resolution deformation result of ROI. The high-spatio-temporal-resolution time-series and/or multidimensional deformation of the ROI are calculated using advanced TS–InSAR technologies, such as PS, SBAS, IPTA, and the multidimensional small-baseline subset (MSBAS) [40–42].

(4) We generate the variable-scale deformation product, combining the high-spatio-temporal-resolution results of ROI and wide-area deformation rate to generate the variable-scale deformation product, which can describe deformation in stable areas only with low-spatial-resolution deformation rate, and in the ROI with the high-spatio-temporal-resolution deformation rate and time series.

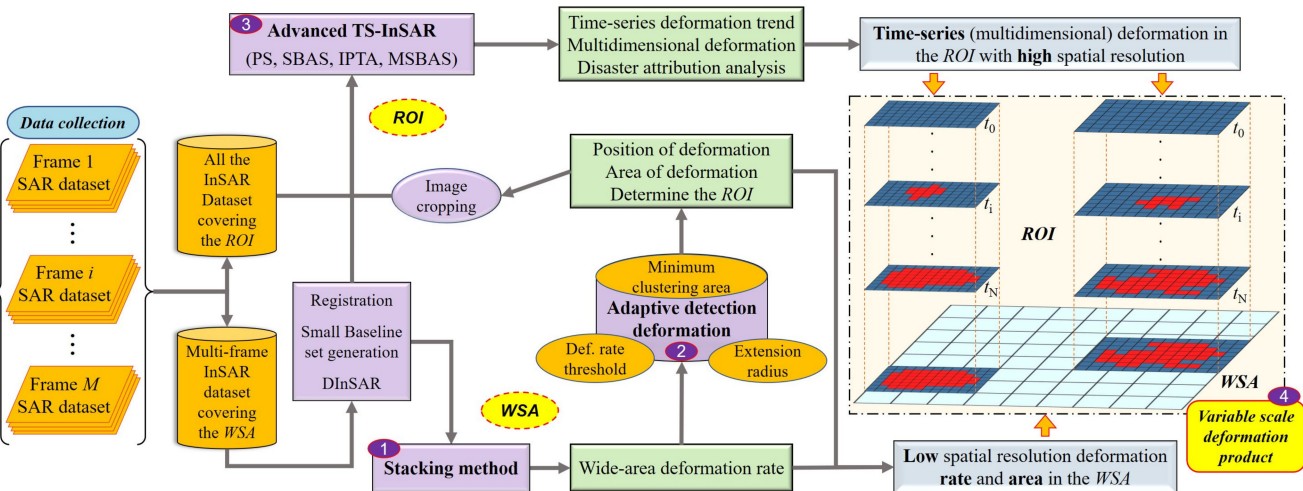

**Figure 1.** Flowchart of the method for the variable-scale monitoring of deformation in a wide area.

### 2.1. Wide-Area Deformation Monitoring Using Stacking

The stacking technology can calculate the deformation rate based on weight and average the unwrapped phases of the multitemporal DInSAR dataset. The stacking technology assumes linear ground-deformation changes, and temporal randomly distributed phase noise, such as atmospheric delay phase. Assuming $N + 1$ SAR images of one frame covering the WSA constitute $M$ InSAR pairs, the displacement phase can be separated as

$$\overline{\phi} = \sum_{i=1}^{M} \phi_i \cdot \Delta t_i \Big/ \sum_{i=1}^{M} \Delta t_i^2 \tag{1}$$

in which $\overline{\phi}$ is the rate of deformation phase change. $\phi_i$ and $\Delta t_i$ are the interference phase and the time interval of the $i$-th InSAR pair, respectively.

The rate of the deformation phase change would be converted to the deformation rate ($V_{def}$) as,

$$V_{def} = \lambda \cdot \overline{\phi}/4\pi \tag{2}$$

where $\lambda$ is the wavelength of the SAR sensor. The multitemporal DInSAR data in each frame is processed using stacking technology to obtain the ground deformation of the WSA.

### 2.2. ROI Detection Based on Wide-Area Deformation Rate

Luo et al. [39] proposed an improved method to automatically identify and evaluate geological hazards using TS–InSAR results. By judging and analyzing the deformation rate and time series in the monitoring area, the method can automatically identify the deformation region and evaluate its hazard grade. In this study, we improve this method to accurately delineate the ROI.

To improve the accuracy of ROI detection, we first apply spatial domain filtering to the wide-area monitoring results to obtain deformation results with good spatial consistency. Then, we set the thresholds for deformation rate, extension radius, and minimum clustering area. When the absolute value of the deformation rate is greater than the deformation rate threshold, it is considered to be an active point. Otherwise, it is a stable point. Buffer zones are established around the active points according to the extension radius. The active points are clustered following the principle of spatial proximity relationship [43]. The clustering regions are smoothed to refine the boundary. The robust deformation regions and their area are obtained by removing regions smaller than the minimum clustering area. The ROI can be finally located based on spatial clustering and the area of deformation.

A detailed description of the intelligent recognition part of the method can be found in [39]. It should be noted that InSAR can only obtain one-dimensional (1D) deformation along the line-of-sight (LOS) direction of the SAR sensor, so the InSAR data of one geometry is insensitive to the deformation of some regions, especially landslides [44]. To obtain more reliable deformation detection results, we need to use the above method and InSAR data from different observational geometry. The detection results of multitrack InSAR data are taken together as the final deformation regions. Then, we can adaptively determine the ROI and perform fine monitoring.

### 2.3. ROI Deformation Refinement Using Advanced TS–InSAR

When calculating the wide-area deformation rate, we select the InSAR data with the same acquisition time from different frames to facilitate the splicing of the results from different frames and to maintain the consistency of the wide-area deformation rate. To accurately monitor the deformation in the ROI, we first crop the registered InSAR datasets. The cropped datasets are used to obtain the time-series and multidimensional ground deformation of the ROI. Detailed steps are as follows.

(1) Deformation time-series calculation. We process the collected InSAR datasets using TS–InSAR technology, with a smaller multi-looking number (a higher spatial resolution). In this study, we use an improved IPTA method to calculate the deformation time series of the ROI [45,46].

(2) Multidimensional deformation rate/time-series calculation. If the ROI has InSAR data with different observation geometry during the same acquisition time, we can obtain the vertical and horizontal displacements using the MSBAS method.

If multi-sensor and multitemporal InSAR data covering the ROI are available, we can collect all data to analyze the long-term deformation and understand the deformation spatio-temporal evolution features based on the data-overlapping and deformation model [47,48].

### 2.4. Variable-Scale Deformation Product Generation

The low-spatial-resolution deformation rate can be used to detect a stable surface in the WSA, which greatly reduces the task and data volume of wide-area InSAR deformation

monitoring. In addition, we obtain the fine results of the deformation time series with a high spatial resolution of ROI using advanced TS–InSAR technology. A variable-scale deformation product organization structure includes low-spatial-resolution deformation rates in stable areas of the WSA and the high-spatio-temporal-resolution deformation in the ROI. Hence, we superimpose the high-spatio-temporal-resolution deformation at the corresponding regions of the ROI on the wide-area deformation rate results to improve the spatial and temporal dimensions of the deformation in the ROI. At this stage, we can obtain variable-scale deformation products in the WSA, which only contain low-spatial-resolution deformation rates in stable regions, and fine monitoring results in the ROI.

## 3. Study Area and Data Processing

### 3.1. The Turpan–Hami Basin

The Turpan–Hami basin, consisting of the Turpan and the Hami depressions, is an intermountain basin located in northwest China (Figure 2). Since the end of the Early Permian period, the Turpan–Hami basin has developed following the model of "fault-depression foreland". It is a typical faulted basin, with limited sedimentary range, great lateral variation of sedimentary thickness, and multiple depositions and subsidence centers. The geological conditions and active tectonic motion contribute to oil and gas accumulation and make the Turpan–Hami basin the largest coal-derived petroleum-producing basin in China [49]. Moreover, there are many mineral resources in this basin, e.g., coal, iron, and potassium (sodium) saltpeter. It is the world's largest potassium (sodium) saltpeter resource. Aydingkol Lake, located in the middle of the Turpan depression, is the lowest depression in China, 154.31 m below sea level [50]. Centering on Aydingkol Lake, the Turpan depression presents a roughly three-ring shape. The outermost ring has high snow-capped mountains. The middle ring is the Gobi gravel belt. The inner ring is an oasis plain belt, most of which belongs to a piedmont sloping plain, and accumulates a large area of fine soil alluvium. The water in the basin mainly comes from rainfall and meltwater from the surrounding mountains. The Tianshan mountains, e.g., Bogurda Mountain and Harlick Mountain, are in the north of the Turpan depression. The Flaming Mountains fault zone lies nearly east–west in the Turpan depression, between Turpan city and Shanshan county (Figure 2). Weathered material is transported from the Tianshan mountains to the center of the basin by water flow, but is blocked by the Flaming Mountains fault line and accumulates in the northern part of the mountains. The surface water and groundwater from the Tianshan mountains are also blocked by the Flaming Mountains fault line. The head height of the shallow aquifers is raised on both sides of the Flaming Mountains, creating overflow zones and an oasis in these areas.

The Turpan–Hami basin has a typical continental warm temperate desert climate, with abundant heat and extremely little precipitation. It has 3200 h of sunshine in a year. The hydrogeology, climate, and lighting conditions make it an ideal place for growing cantaloupe, grapes, cotton, and off-season vegetables. Groundwater is the main source of agricultural water in the arid area. Previously, karezes were the predominant underground water conservancy project in this region. A karez uses the principle of water potential artware to divert water from shallow aquifers to the surface for irrigation. There are more than 2000 karezes in the Turpan–Hami basin, accounting for more than 70% of the total number of karezes in Xinjiang [51,52]. However, many electromechanical wells have been built in the Turpan–Hami basin since the 1960s. Groundwater exploitation has increased yearly, with the annual overexploitation reaching $2.48 \times 10^{10}$ m$^3$, leading to the continuous decline of groundwater level. Advanced water conservancy facilities have reduced people's dependence on karezes. Meanwhile, the water supply source of karezes is shallow aquifers. The continuous reduction of groundwater level directly leads to the decrease or even drying-up of karezes [53]. The number of water-filled karezes in the Turpan depression decreased from 1237 in 1957 to 214 in 2014 [51]. In addition, the increased demand and excess consumption of water resources in upstream areas have seriously threatened the water supply of Aydingkol Lake, resulting in water area shrinkage. The exploitation

of groundwater and mineral resources will make the surface of the Turpan–Hami basin unstable and threatened by potential geohazards.

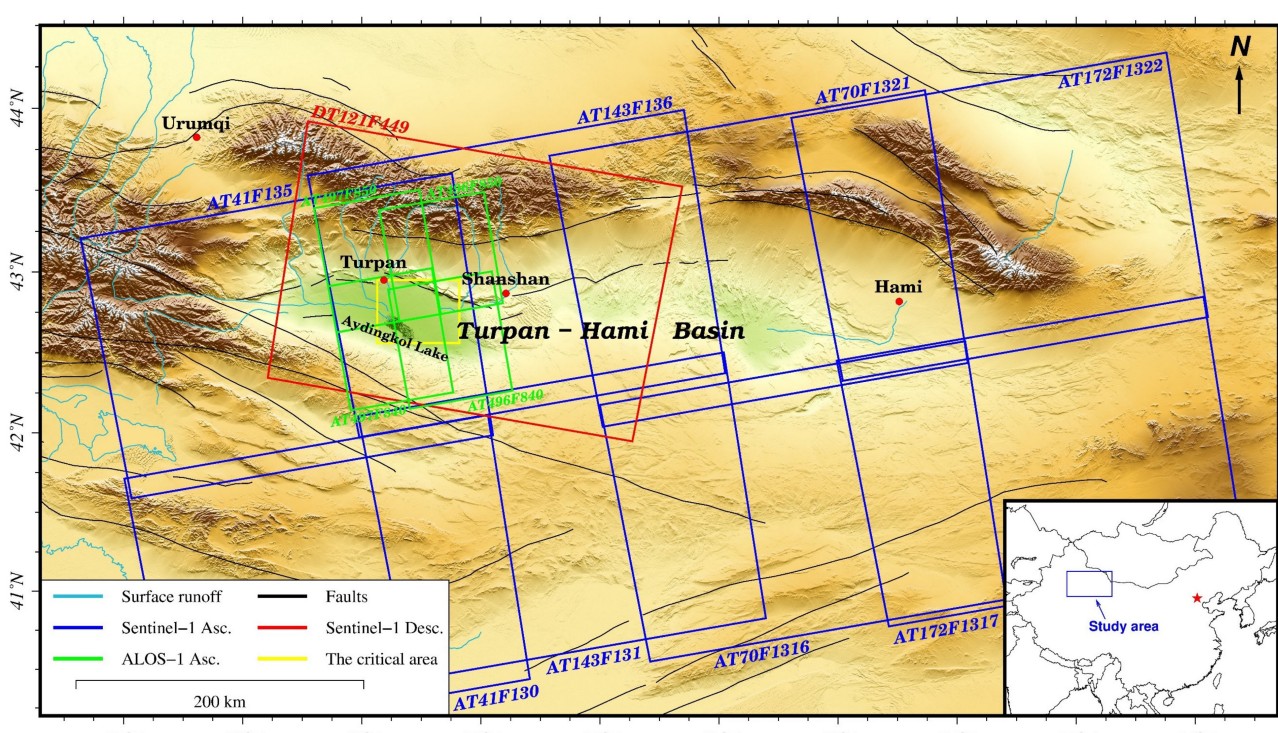

**Figure 2.** The Turpan–Hami basin and SAR data coverage.

### 3.2. InSAR Datasets

To monitor wide-area deformation in the Turpan–Hami basin, we collected eight frames of InSAR data covering the whole Turpan–Hami basin from the Sentinel-1 satellite. The Sentinel-1 satellite began operation in April 2014, and has different observation periods in different regions, resulting in inconsistent periods of SAR data in different regions. To ensure the consistency of deformation rates from multiple frames, we selected the images (628 images in total) from the eight frames acquired from October 2017 to May 2020 (Table 1). The spatial coverage of each dataset is shown in Figure 2.

**Table 1.** Acquisition periods of the datasets.

| Sensor | Frame | Time | Number | Frame | Time | Number |
|---|---|---|---|---|---|---|
| Sentinel-1 | AT172F1317 AT172F1322 | 13/10/2017–30/05/2020 | 77 77 | AT143F131 AT143F136 | 11/10/2007–28/05/2020 | 81 81 |
| | AT70F1316 AT70F1321 | 18/10/2017–30/05/2020 | 78 78 | AT41F130 AT41F135 | 16/10/2017–21/05/2020 | 78 78 |
| | DT121F449 | 19/03/2015–27/04/2020 | 107 | AT41F135 | 25/03/2015–21/05/2020 | 123 |
| ALOS-1 | AT496F840 AT496F850 | 22/01/2007–14/09/2009 | 11 11 | AT497F840 AT497F850 | 08/02/2007–04/10/2010 | 11 11 |

Wide-area InSAR deformation shows that many subsidence funnels are concentrated in the south part of the Flaming Mountains fault zone in the Turpan depression (hereafter referred to as the SFM–def region). The SFM–def region (the yellow box in Figure 2) was selected as an application demonstration area of ROI to carry out the fine monitoring of the deformation time series. Four frames from the ALOS-1/PALSAR dataset spanning from 2007 to 2010 (green rectangles in Figure 2) and a descending track from the Sentinel-1 dataset (red rectangle in Figure 2) covering the SFM–def region were collected. The common

monitoring time of the Sentinel-1 ascending (AT41F135) and descending (DT121F449) tracks data is from 2015 to 2020 (Table 1). These data were used to precisely monitor the long-term and fine deformation in the SFM–def region.

### 3.3. Data Processing

We preprocessed all InSAR datasets covering the WSA. In each frame, one image was selected as the master image to register and resample the rest images. Multitemporal InSAR pairs were generated from SAR data in the same frame, based on the appropriate spatio-temporal baseline thresholds. All multitemporal DInSAR pairs were processed using GAMMA software [54] and two-pass DInSAR technology [55] to obtain multitemporal deformation signals. The shuttle radar topography mission (SRTM) digital elevation model (DEM) with a resolution of 30 m [56] was employed to remove the topographic phases. The point targets with a coherence lower than 0.3 were eliminated [57]. Least-squares-based filtering and the minimum cost flow method [58] were then applied to further suppress phase noise [59] and unwrap the differential interferogram, respectively.

The eight frames of the Sentinel-1 data were preprocessed with a spatial baseline (perpendicular) and temporal baseline of 100 m and 48 days, respectively, and a multi-looking operation of 20:4. Then, stacking was used to process all the multitemporal deformation signals in each frame, to obtain a wide-area deformation rate of the Turpan–Hami basin. The adaptive deformation detection method proposed in Section 2.2 was used to delineate the deformation regions. The thresholds of the deformation rate, extension radius, and minimum clustering area were set as $\pm 2$ cm/year, 250 m, and 1 km$^2$, respectively.

For the SFM–def region, we set the multi-looking parameters of ALOS-1/PALSAR and Sentinel-1 data as 3:8 and 8:2, respectively. The improved IPTA method was used to compute the four frames of the ALOS-1/PALSAR data and the ascending/descending tracks from the Sentinel-1 datasets to obtain long-term and high-resolution displacements. Moreover, MSBAS technology was used to obtain multidimensional deformation from the ascending/descending tracks of the Sentinel-1 datasets. Then, we obtained the variable-scale deformation product of the Turpan–Hami basin, which consists of low-spatial-resolution deformation rates in the stable areas and high-spatio-temporal-resolution deformation in the SFM–def region.

## 4. Results

### 4.1. Monitoring and Detecting the Wide-Area Deformation in the Turpan–Hami Basin

The wide-area ground subsidence in the Turpan–Hami basin (Figure 3) shows that the surface of the Turpan–Hami basin is generally stable. The regions with deformation account for a small proportion of the whole. The main deformation type is subsidence. Based on the deformation detection threshold set in Section 3.3, we identified 32 deformation areas (the funnel) in the Turpan–Hami basin (the blue lines in Figure 3). The area of each funnel is shown in Table 2. The detected deformation area accounts for about 2.4‰ of the total monitoring area.

Analyzing the hydrogeology and land cover of the deformation areas, we divided the ground deformation in the Turpan–Hami basin into three types:

(1) Ground subsidence in agricultural areas caused by groundwater overexploitation. This kind of subsidence has the largest area and is concentrated in the oasis plain south of the Flaming Mountains fault zone (Figure 3a).

(2) Ground subsidence associated with mineral mining. This kind of deformation is sporadically distributed over the Turpan–Hami basin. Such deformation regions have a small area but large deformation magnitude, e.g., Figure 3b.

(3) Ground uplift associated with the lake water withdrawal, resulting in saline–alkali lands. This kind of deformation is mainly distributed around Aydingkol Lake, characterized by small magnitude and mainly horizontal movement (Figure 4e,f).

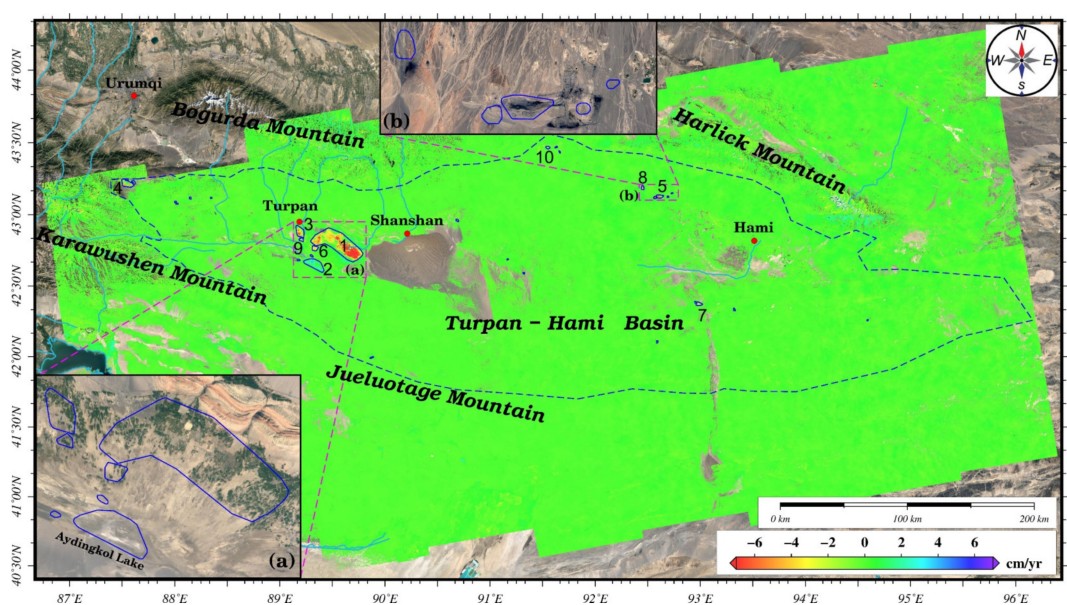

**Figure 3.** Wide-area subsidence rate map and the detected deformation regions. The numbers identify the location of the top 10 deformation regions. (**a**) The SFM–def region in Figure 4. (**b**) One of the major mining areas. Background image: Google Maps satellite image.

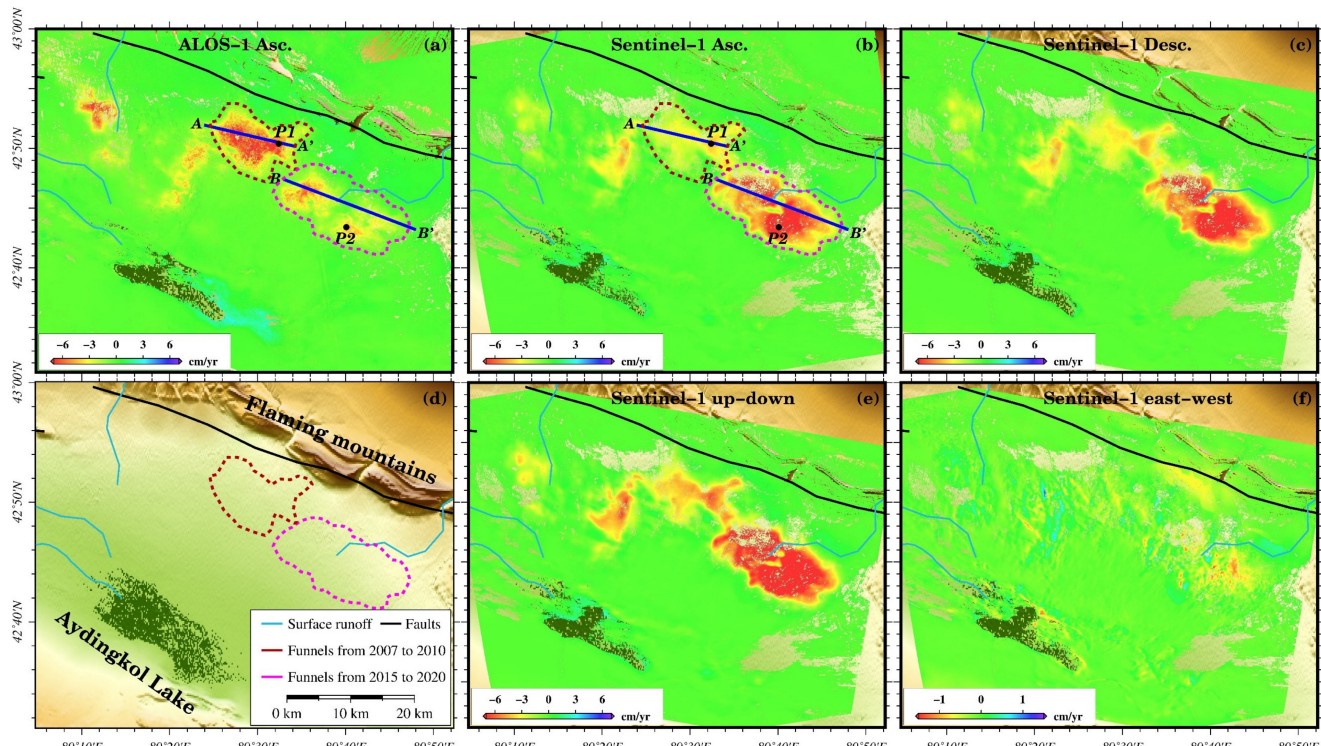

**Figure 4.** Deformation rate along LOS directions from (**a**) ALOS-1/PALSAR, (**b**) ascending track Sentinel-1, and (**c**) descending track Sentinel-1 data. Negative values indicate the direction away from the SAR satellite, while positive values indicate the opposite. (**d**) The hydrogeology of this area. (**e**,**f**) The deformation rate along the up–down and east–west directions calculated from ascending/descending tracks Sentinel-1 data. The red dotted line delineates the central area of the subsidence funnels from 2007 to 2010. The magenta dotted line delineates the central area of the subsidence funnels from 2015 to 2020.

**Table 2.** The area of the deformation funnels.

| Num. | Area (km²) | Num. | Area (km²) | Num. | Area (km²) | Num. | Area (km²) |
|---|---|---|---|---|---|---|---|
| 1 | 437.6 | 9 | 5.5 | 17 | 2.3 | 25 | 1.6 |
| 2 | 61.2 | 10 | 5.1 | 18 | 2.3 | 26 | 1.5 |
| 3 | 42.6 | 11 | 4.8 | 19 | 2.2 | 27 | 1.5 |
| 4 | 26.1 | 12 | 4.1 | 20 | 2.2 | 28 | 1.5 |
| 5 | 16.1 | 13 | 3.4 | 21 | 2.1 | 29 | 1.4 |
| 6 | 11.1 | 14 | 3.1 | 22 | 2.1 | 30 | 1.2 |
| 7 | 9.1 | 15 | 2.6 | 23 | 2.0 | 31 | 1.1 |
| 8 | 6.8 | 16 | 2.5 | 24 | 1.9 | 32 | 1.0 |
| Total area (km²) | | | | 669.6 | | | |

The largest deformation funnel is distributed in the SFM–def region, with an area of 437.6 km², surrounded by small funnels (Figure 3a). The optical images show that the subsidence funnels in the SFM–def region are highly correlated with the location of agricultural areas. Aydingkol Lake is in the south of the SFM–def region (Figure 3a). In recent years, the area of the lake has continuously shrunk, and a large area of saline–alkali land has appeared. There is obvious ground uplift in these saline–alkali regions. In addition, multiple subsidence funnels are observed close to some mines, e.g., the funnel cluster in Figure 3b. The wide-area deformation results are discussed in detail in Section 5.

### 4.2. Deformation Time Series of the SFM–Def Region from 2007 to 2020

#### 4.2.1. Long-Term Deformation in the Spatial Dimension

The long-term (2007–2010 and 2015–2020), multidimensional (along with up–down and east–west directions), and high-spatial-resolution displacements are obtained from the four frames of the ALOS-1/PALSAR data and the ascending/descending tracks from the Sentinel-1 data, using advanced IPTA and MSBAS technologies (Figure 4). The ground deformation is mainly distributed in a plain area south of the Flaming Mountains fault line (Figure 4d). There are large areas of farmland in this region (Figure 3a), and the irrigation relies heavily on groundwater. The ground deformation is mainly vertical, with small horizontal movement (Figure 4e,f), which is typical for displacements caused by groundwater extraction [60–62]. The red and magenta dotted lines in Figure 4 delineate the settlement funnel centers in 2007–2010 and 2015–2020, respectively. The area and magnitude of the subsidence in the northwest of the SFM–def region gradually decrease, but the subsidence area in the southeast gradually expands and becomes connected. The center of the funnel shifts from the northwest to the southeast, and form a giant funnel with a larger subsidence rate and area in the southeast region. See Section 5 for detailed analysis and discussion.

#### 4.2.2. Long-Term Deformation in the Time Dimension

The long-term deformation rate can reflect the spatial distribution and evolution characteristics of ground deformation. We select two profiles (AA′ and BB′) and two points (P1 and P2) in the SFM–def region (Figure 4) to investigate the variation characteristics of deformation in the time domain. The long-term time-series displacements at the corresponding position in the two monitoring periods, i.e., 2007–2010 and 2015–2020, are shown in Figures 5 and 6. The time-series cumulative deformation at AA′ and P1, BB′ and P2 can represent the deformation characteristics of the central region of the subsidence funnels during the two monitoring periods.

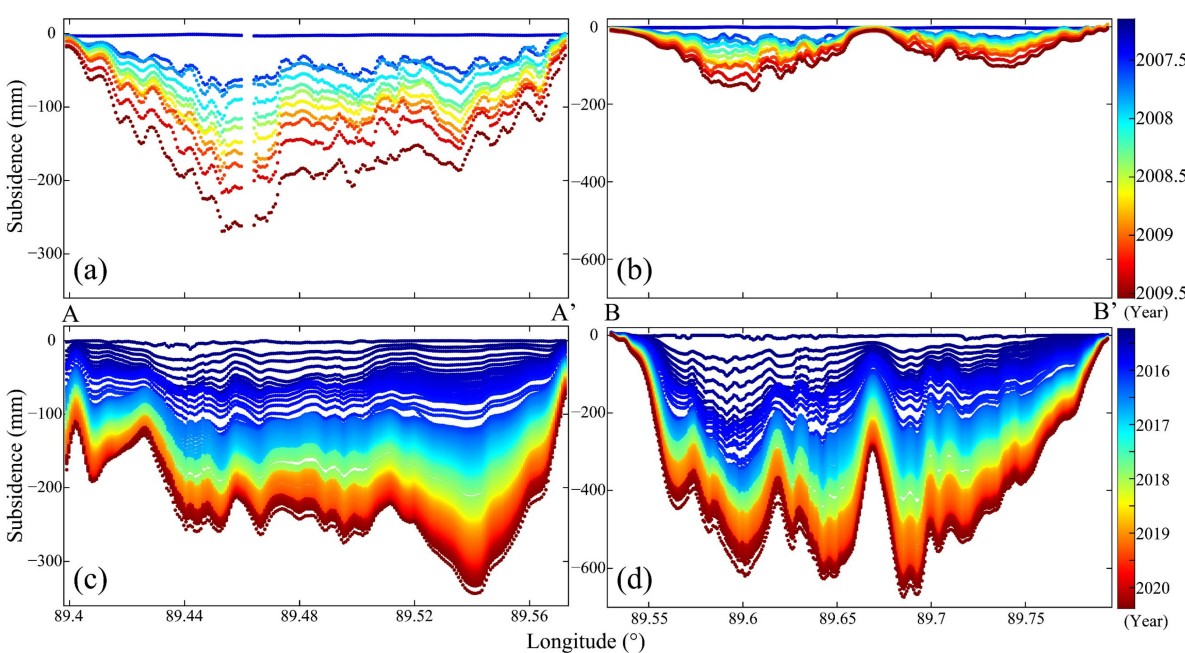

**Figure 5.** The long-term cumulative deformation at the profiles (**a**,**c**) A-A′ and (**b**,**d**) B-B′ in Figure 4.

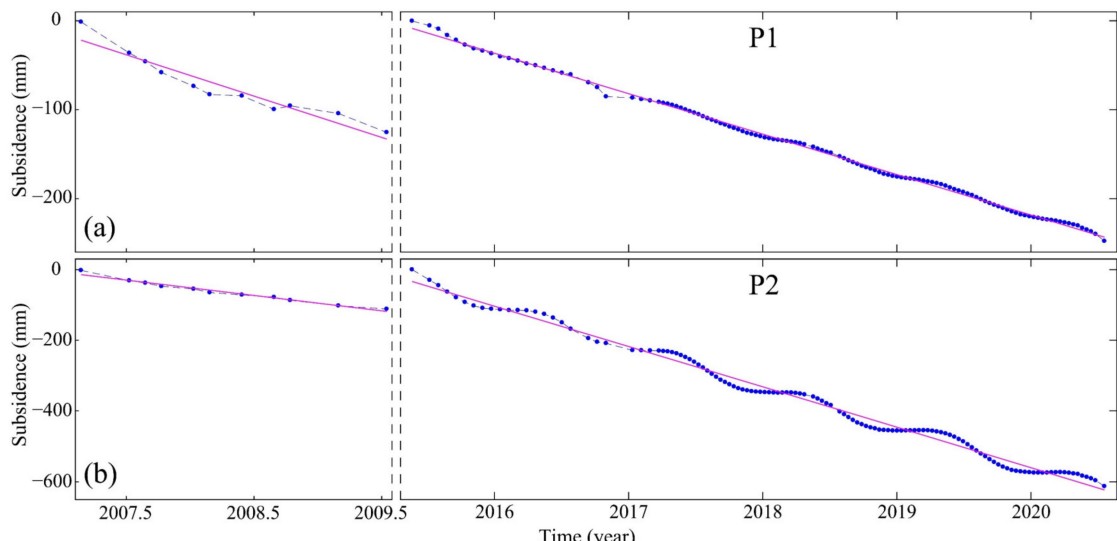

**Figure 6.** The time-series cumulative deformation at P1 and P2 in Figure 4. The blue dots represent the InSAR observations. The magenta lines are the linear fitting results of the corresponding InSAR observations.

The long-term deformation at AA′ and P1 shows that the subsidence of the area with the most significant subsidence in the first monitoring period tends to be stable, and slows down in the second period. In the first period, the subsidence rate of the section northwest of AA′ is higher than that of the southeast section. However, in the second period, this phenomenon is reversed. The subsidence center moves from northwest to southeast, which is consistent with the spatial evolution of the global subsidence funnel. In the first period, the subsidence rate of BB′ is small, and presents two separate funnels. In the latter period, the two funnels merge into a giant funnel. The subsidence area and rate increases significantly.

Both ALOS-1/PALSAR and Sentinel-1 data can reflect the overall change characteristics of the subsidence in time and space well (Figures 4 and 5). However, compared with

the ALOS-1/PALSAR data, which have a revisit period of ≥46 days, the Sentinel-1 data can capture more detailed changes to the deformation signals with obvious periodicities in the time dimension due to its higher temporal resolution (≥12 days) (Figure 6). The subsidence mainly occurs in summer. The ground tends to be stable or slightly uplifted in winter. See Section 5 for detailed analysis and discussion.

### 4.2.3. Reliability Assessment

As can be seen from Figure 4, the deformation results of the SFM–def region from the ALOS-1 data of different frames have good consistency. The deformation obtained by the Sentinel-1 data of ascending and descending tracks also has good consistency in spatial distribution and magnitude. This indicates that the TS–InSAR results have a good consistency. To quantitatively assess the reliability of the TS–InSAR results, we compare the average subsidence rates extracted from the overlapped areas of two adjacent InSAR frames acquired at the same period, e.g., ALOS-1/PALSAR datasets from AT496F840 and AT496F850, and Sentinel-1 datasets from AT143F136 and DT121F449 (Figure 2). Due to the different observation geometry of each monitoring point in different frames, we convert the LOS deformation to the vertical direction for comparison. The correlation between the results at AT496F840 and AT496F850, and Sentinel-1 results at AT143F136 and DT121F449, are 0.98 and 0.99, respectively. The root-mean-square errors (RMSEs) between them are 0.02 and 0.01 mm/year, respectively. These results show good consistency, and the differences at most points are smaller than three times the RMSE (between the red dotted lines in Figure 7).

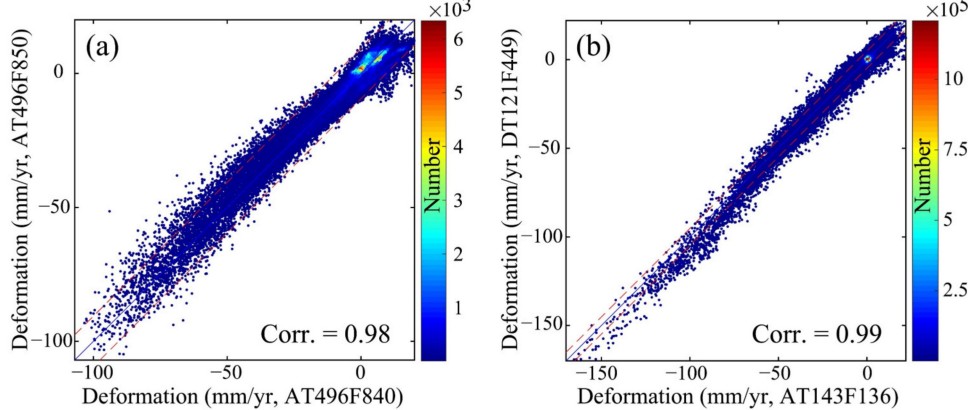

**Figure 7.** Comparison between the results obtained by (**a**) ALOS-1/PALSAR AT496F840 and AT496F850 data, and (**b**) the Sentinel-1 AT143F136 and DT121F449 data. The red dotted lines denote the value three times the root-mean-square error.

## 5. Discussion

### 5.1. Anthropogenic Factors of Ground Deformation in the Turpan–Hami Basin

We obtained variable-scale deformation products in the Turpan–Hami basin using the proposed WAVS–InSAR method. The distribution of most detected deformation funnels (Section 4.1) is highly consistent with human activity, such as agriculture cultivation and mineral mining. The agricultural area in the SFM–def region has a funnel cluster with the largest deformation area and magnitude in the Turpan–Hami basin. We obtained the long-term and multidimensional ground deformation in the SFM–def region in Section 4.2. The subsidence center of the first period (2007–2010) shifted from the northwest to the southeast in the second period (2015–2020).

We collect optical images of the SFM–def region in 2007 and 2018 (Figure 8a,b), corresponding to the two monitoring periods. The green lines mark the locations of the greenhouses that appeared in the latter period. As the optical images show, the majority of farmland in the SFM–def region in 2007 was open-air farmland. However, in 2018, there was a large area of greenhouses, especially in the farmland far from the Flaming Mountains

fault line. Many open-air farmlands in 2007 had been changed to greenhouses (Figure 8). In traditional open-air farmland, the crops are mainly grain and cotton, which are planted in spring, managed in summer, and harvested in autumn. However, in greenhouse farmland, the expected proportion of fruit and vegetable cultivation is more than 70% [63]. After 2009, many greenhouses were built in Turpan, especially in the agricultural areas far from the southern margin of the Flaming Mountains fault line (Figure 8b). Advanced agricultural planting technologies have brought huge economic benefits to Turpan, but also increased the environmental burden, especially the demand for water [53]. Irrigation water in the SFM–def region is mainly groundwater. Hence, ground subsidence caused by groundwater overexploitation is more significant in the greenhouse areas of the SFM–def region, resulting in aquifers carrying net deficit and the subsidence center shifting to the southeast (Figure 8a,b).

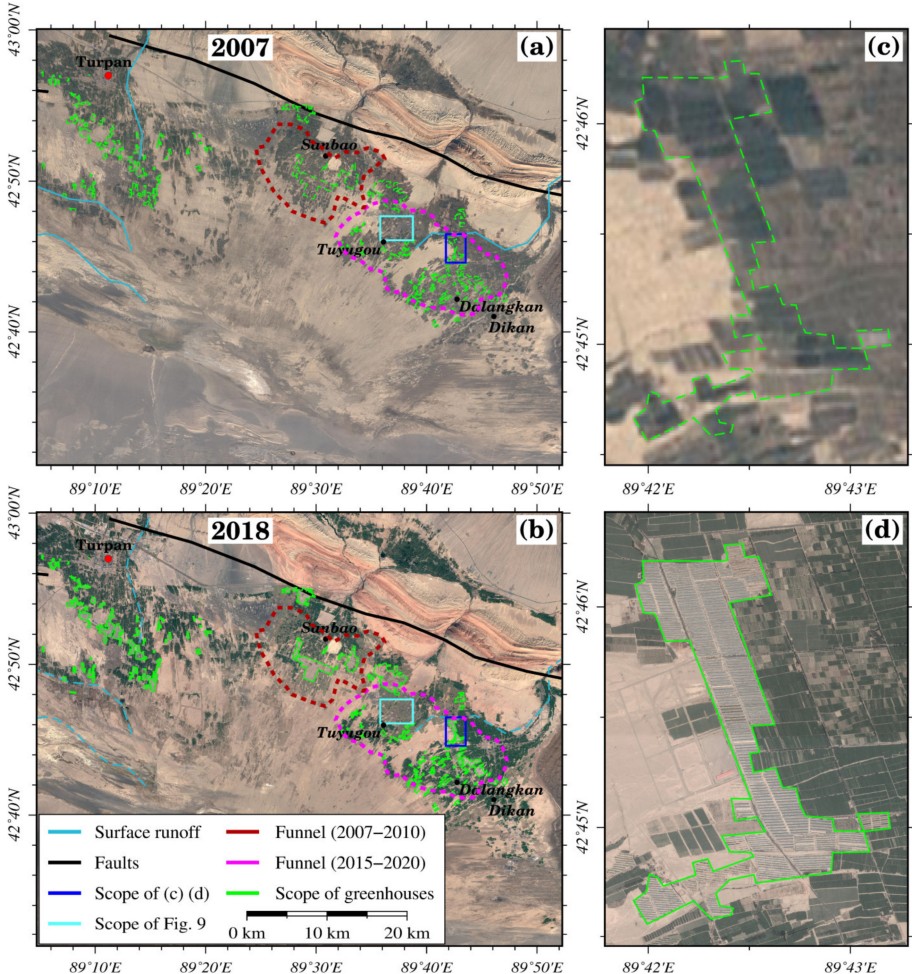

**Figure 8.** (**a**,**b**) Optical images of the SFM–def region in 2007 and 2018. Green lines delineate greenhouse planting areas. (**c**,**d**) Zoom-ins of the blue rectangular area in (**a**,**b**) in 2007 and 2018. Background image: Google Maps satellite image.

Karezes are an important water supply in arid agricultural areas, known as "the fountains of life". In China, karezes are mainly distributed in the Turpan–Hami basin (Section 3.1). A karez is composed of vertical shafts, culverts, water outlets, open channels, and waterlogging dams, with length ranging from several to dozens of kilometers [51]. The number and distribution of karezes can reflect the changes to the ecological environment in the Turpan–Hami basin. It is important to evaluate the health of the aquifer. We compared two high-resolution (0.44 m) optical images covering the blue rectangular region of Figure 8a,b in July 2003 and May 2013 (Figure 9), where ground subsidence funnels in the

second period (Figure 9a,b) were developed. In July 2003, lots of small mounds—the shaft part of a karez—are linearly distributed in this area (Figure 9c). However, a lot of small mounds have disappeared in the optical image taken in May 2013, indicating the karezes in the area were severely damaged (Figure 9d). Some areas that karezes passed through were turned into farmland (Figure 9e,f). The water supply of karezes was destroyed. The water supply in this area will depend mainly on the extraction of groundwater using electromechanical wells. The karezes may have ceased to function, and dried up.

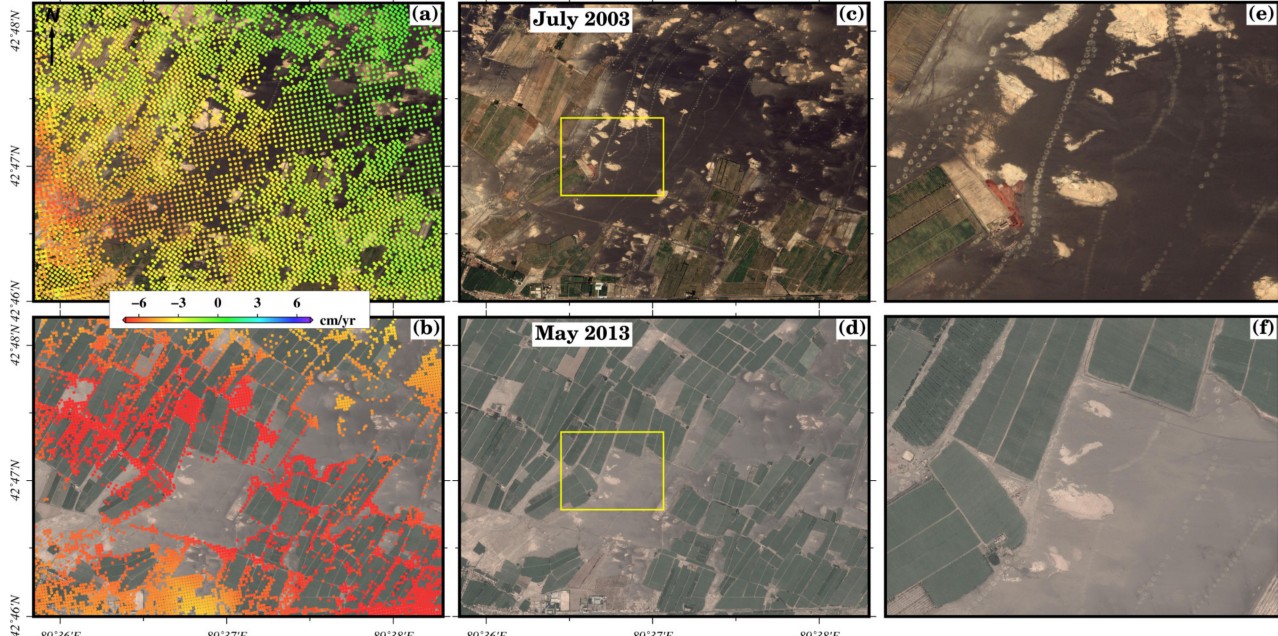

**Figure 9.** Deformation rate in the blue rectangular area of Figure 8 from (**a**) ALOS-1/PALSAR, (**b**) ascending track Sentinel-1 data. (**c**,**d**) Optical images of this area in July 2003 and May 2013, respectively. (**e**,**f**) Zoom-ins of the yellow rectangular area in (**c**,**d**). Background image: Google Maps satellite image.

In addition, we collected land cover data of the SFM–def region in 2000, 2010, and 2020 (Figure 10) (data from global Land Cover Data Product and Service website of National Basic Geographic Information Center of China (http://www.globallandcover.com/, accessed on 10 July 2022)), and the corresponding area of land cover type in each period (Table 3). The agricultural area has continuously expanded in the past two decades. Artificial areas have expanded rapidly in the past decade, more than 10 times the rate of the previous decade. Water and wetland areas have decreased in the last decade. In 2000 and 2010, the lake area and the surrounding wetland area of Aydingkol Lake was stable, indicating that surface runoff and groundwater are still effective for supply of the lake. These water sources can also partially alleviate the overexploitation of groundwater for agricultural use. However, in 2020, land cover data showed that the waters and wetland of Aydingkol Lake had almost disappeared. Farmland area is in the inner ring of Aydingkol Lake (Section 3.1). The excessive use of surface and underground water in farmland areas has seriously reduced the water supply of the lake, resulting in the shrinkage of water and wetland, which will seriously endanger the ecological environment. The transformation of local agriculture and the economy has upset the ecological balance in the SFM–def region and the balance of aquifers. Conflicts between the development of the local agricultural economy and ecological environment should arouse the attention of local governments.

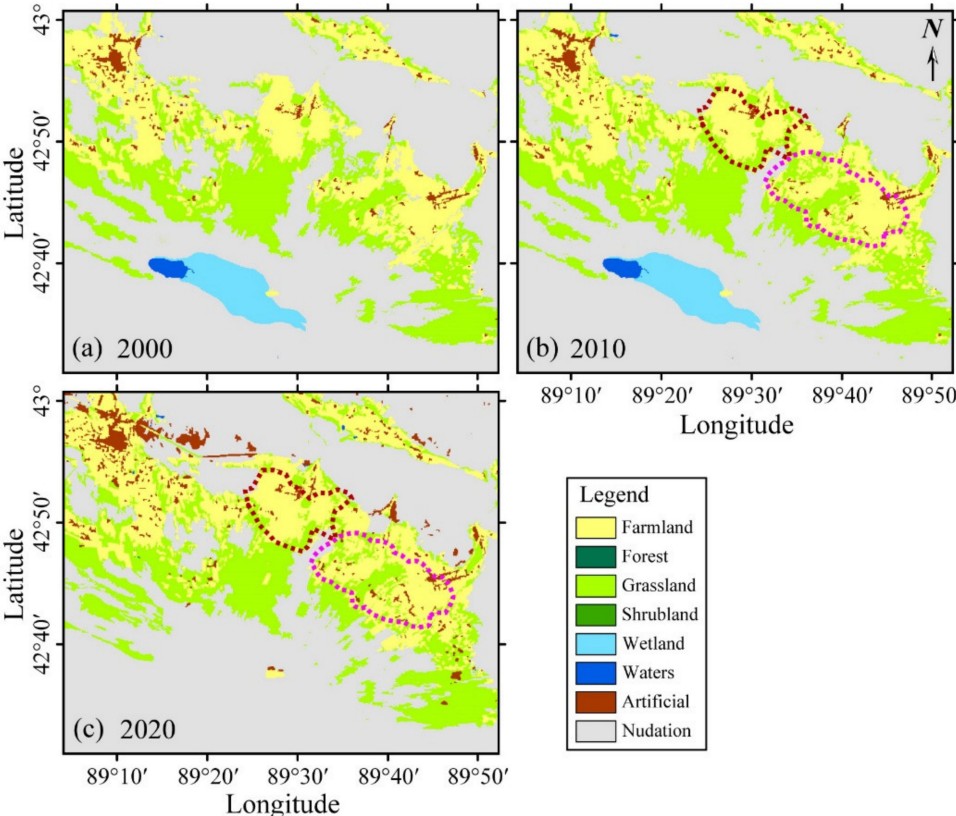

**Figure 10.** Spatio-temporal evolution of the land covers in the SFM–def region in 2000, 2010, and 2020. The red and magenta dotted lines delineate the central area of the subsidence funnels during the periods 2007–2010 and 2015–2020, respectively.

**Table 3.** The area of different land covers in the SFM–def region in 2000, 2010, and 2020, obtained from Globeland 30.

| Sort<br>Time | Farmland | Grassland | Wetland | Waters | Artificial | Nudation |
|---|---|---|---|---|---|---|
| 2000 | 574.33 | 652.74 | 99.90 | 13.68 | 43.81 | 2288.24 |
| 2010 | 657.46 | 669.62 | 99.90 | 14.09 | 48.50 | 2181.47 |
| 2020 | 705.65 | 658.64 | 0.16 | 1.53 | 110.14 | 2194.59 |
| Percentage 1 [a] | 14.5% | 2.6% | 0 | 3.0% | 10.7% | −4.7% |
| Percentage 2 [b] | 7.3% | −1.6% | −99.8% | −89.1% | 127.1% | 0.6% |
| Percentage 3 [c] | 22.9% | 0.9% | −99.8% | −88.8% | 151.4% | −4.1% |

Unit: km[2]. [a]: Percentage of numerical growth in 2010 compared with 2000. [b]: Percentage of numerical growth in 2020 over 2010. [c]: Percentage of numerical growth in 2020 compared with 2000.

### 5.2. Geological Explanation of Ground Deformation in the Turpan–Hami Basin

There are many farmlands in both the Turpan and Hami depressions. Facility agriculture planting areas are also developed in other agricultural areas, e.g., the oasis areas in Hami and the western part of Turpan. However, why is there a large area of ground subsidence funnels in only the agricultural areas of the SFM–def region?

We plotted the deformation results and the corresponding optical images and faults of the oasis areas in the Turpan depression and the Hami depression (Figure 11). Rainfall is scarce in the Turpan–Hami basin. Irrigation water in the oasis agricultural areas depends on rainfall and meltwater from the surrounding mountains (Section 3.1). The Flaming Mountains fault line lies east–west in the Turpan depression, blocking water flowing

from the Tianshan mountain to the south. The other areas, e.g., the northern part of the Flaming Mountains fault line and Hami, can directly obtain abundant mountain water. The surplus water in the Hami oasis can even form a river to supply the downstream area in the southwest (Figure 11d). However, the SFM–def region is short of surface water and groundwater, and the only river channel has almost dried up. As the distance from the southern margin of the fault increases, the water supply gradually decreases. The limited surface water cannot meet the continuously increasing demand for irrigation water, resulting in the continuous overexploitation of aquifers, causing the development of many subsidence funnels in this area.

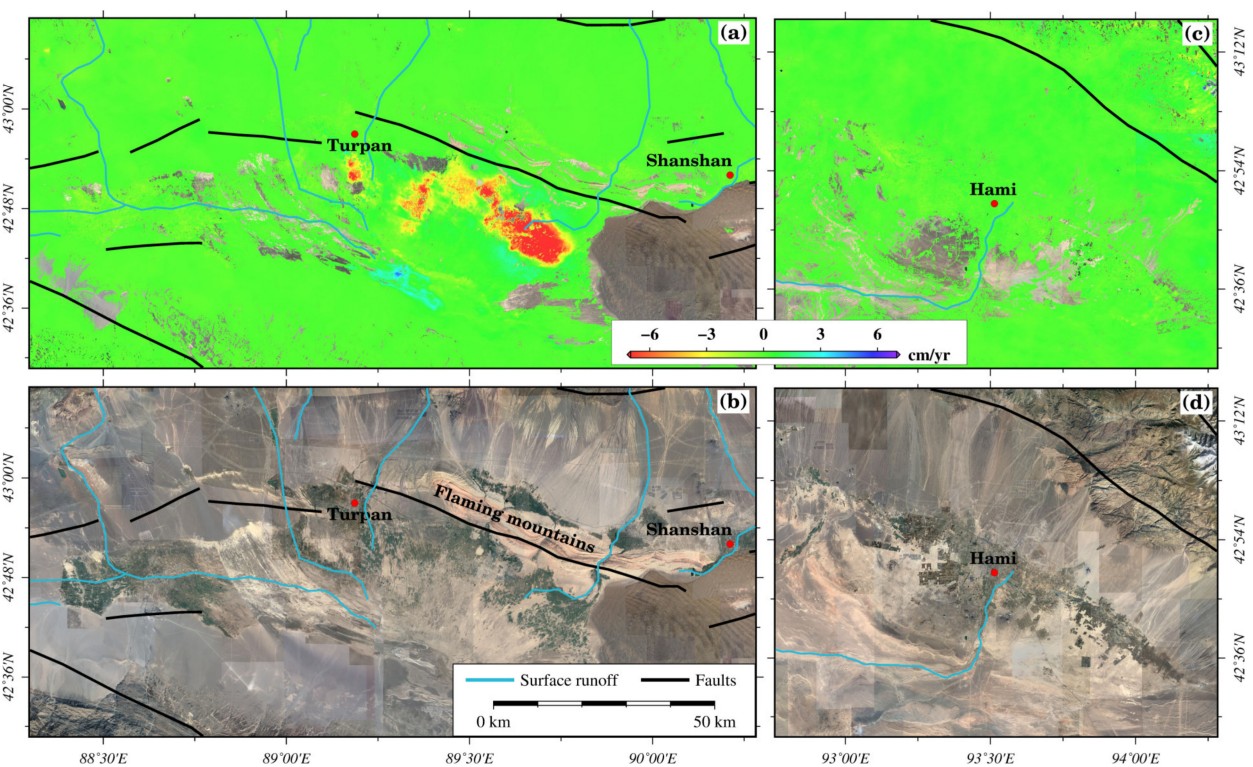

**Figure 11.** Comparison of ground deformation and optical images in (**a**,**b**) Turpan and (**c**,**d**) Hami oases. Background image: Google Maps satellite image.

The climate in the Turpan–Hami basin is dry and sunny, so evaporation is serious, especially in late spring and summer, when 75% of the year's evaporation occurs. Summer is also the main period of crop growth, which demands more water for irrigation. The agriculture in the SFM–def region relies heavily on groundwater exploitation, which directly leads to the short-term sharp loss of aquifers, and accelerates surface subsidence. From late autumn to early spring, groundwater exploitation intensity in farmland decreases. The aquifers are replenished by surface runoff and groundwater reflux. This explains why the subsidence of the funnels in farmland accelerates in summer and autumn, and slows down or turns to slight uplift in winter and spring (Figure 6).

### 5.3. Development of InSAR Deformation Monitoring in a Wide Area

At present, most wide-area InSAR deformation monitoring projects use the TS–InSAR algorithm to resolve the deformation time series of all highly coherent monitoring points in each frame [10,11,13,17]. Even though different multi-looking ratios for the WSA and the ROI were used to improve the efficiency of data processing by controlling the spatial resolution of the results [14], these methods are still not out of the scope of time-series deformation calculation. Moreover, if the strategy of reducing spatial resolution is not optimized, it will cause repeated calculations and reduce monitoring efficiency. The deformation rate is usually used to detect potential geohazards in a wide area [38]. Therefore, the deformation

time series of some points is unnecessary, especially for stable areas. Hence, calculating the deformation time series at all monitoring points wastes computing resources and labor costs, and produces lots of redundant results. For example, deformation areas account for about 2.4‰ of the total monitoring area in the Turpan–Hami basin. WAVS–InSAR only calculates the deformation rate at each monitoring point in the WSA. Reducing the time dimension of the wide-area deformation results can greatly improve the efficiency of the multitemporal InSAR solution, especially for a lot of InSAR frames in the WSA. Spatial distribution and area of deformation are detected by an adaptive deformation detection method combined with the obtained wide-area deformation rate. After that, high-precision time-series monitoring is only done in the ROI to obtain effective fine deformation results.

For variable-scale deformation results, the WAVS–InSAR strategy proposes a novel variable-scale deformation product organization structure, i.e., it shows the deformation information at the stable surface with low-spatial-resolution deformation rate, while the ROI has a high-spatio-temporal-resolution deformation time series. This structure reduces the amount of deformation results in the stable regions of the WSA, locates the ROI efficiently, and improves the spatial and temporal dimensions of the deformation in the ROI, which is convenient for the calculation, storage, display, and interpretation of the deformation results.

As the SAR satellites and InSAR data increase, InSAR deformation monitoring projects will produce many monitoring results. In the future, wide-area InSAR deformation monitoring projects should be object-oriented, integrating different deformation monitoring to obtain deformation results of multidimensional and high spatio-temporal resolution, and ultimately form a set of universal deformation products. The data-processing strategy and deformation product organization structure proposed in WAVS–InSAR will greatly improve deformation monitoring efficiency and reduce the storage space of massive InSAR monitoring data, which may become a standardized data-processing procedure and data-storage format for future wide-area InSAR deformation products.

## 6. Conclusions

In this study, we proposed a variable-scale InSAR ground-deformation detection strategy and a deformation product organization structure for wide-area monitoring, namely WAVS–InSAR. This strategy efficiently obtains the deformation rate in the WSA, and uses an adaptive deformation detection method to process the wide-area deformation rate and obtain the spatial distribution and area of the deformation areas (ROI). High-precision time-series monitoring is then only done in the ROI, to obtain effective fine deformation results. Therefore, we can produce variable-scale deformation products in the WSA that consist of low-spatial-resolution deformation rates in stable regions, and fine monitoring results in the ROI.

The proposed WAVS–InSAR was used to monitor wide-area deformation in the Turpan–Hami basin, which has an area of 277,000 km$^2$. The results show that there are 32 deformation regions with an area of more than 1 km$^2$ and a deformation magnitude of more than 2 cm/year. The detected deformation areas account for about 2.4‰ of the total monitoring area. The SFM–def region is selected as an application demonstration area of the ROI to carry out fine monitoring of the deformation time series. We obtain the long-term and multidimensional deformation of this area from 2007 to 2010 and from 2015 to 2020 using improved IPTA and MSBAS technologies.

The subsidence funnel center in the SFM–def region moved from northwest to southeast during 2007 to 2020. Based on the variable-scale deformation products and the information regarding hydrogeology, land cover and human activities, we analyze the causes of ground subsidence. Tectonic faults have blocked the water supply in the SFM–def region. The rapid development of facility agriculture has increased the water demand for irrigation. To solve this problem, groundwater has been overexploited. The aquifers in the oasis plain in the SFM–def region are in a state of net deficit. Increased demand for water in the upper reaches of Aydingkol Lake has reduced the lake's water supply. Aydingkol Lake has

shrunk dramatically. In addition, there are several deformation areas related to mining in the Turpan–Hami basin.

**Author Contributions:** Conceptualization, Y.W., G.F. and Z.L.; Formal analysis, H.W. and Z.X.; Funding acquisition, Y.W., G.F. and Z.L.; Investigation, J.Z. and J.H.; Methodology, Y.W. and G.F.; Resources, G.F. and Z.L.; Software, Y.W.; Supervision, Z.L., J.Z. and J.H.; Validation, Y.W., S.L. and H.W.; Writing—original draft, Y.W. and G.F.; Writing—review and editing, all authors. All authors have read and agreed to the published version of the manuscript.

**Funding:** This research was funded by the Natural Science Foundation of Hunan Province (2021JJ30807), the National Natural Science Foundation of China (Nos. 42174039), the National Science Fund for Distinguished Young Scholars (41925016), the Scientific Research Innovation Project for Graduate Students in Hunan Province (CX20200111), and the Fundamental Research Funds for the Central Universities of Central South University (2020zzts168).

**Data Availability Statement:** The data used to support the findings of this study are available from the corresponding author upon request.

**Acknowledgments:** The authors would like to thank the European Space Agency (ESA) for providing free and open Sentinel-1 data, the Japan Aerospace Exploration Agency (JAXA) for providing the ALOS-1/PALSAR images (No: PER2A2N038), and the National Basic Geographic Information Center of China for providing the free and open global Land Cover Data Product (http://www.globallandcover.com/, accessed on 10 July 2022).

**Conflicts of Interest:** The authors declare no conflict of interest.

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
