# Peer review of "A Strategy for Variable-Scale InSAR Deformation Monitoring in a Wide Area: A Case Study in the Turpan–Hami Basin, China"

_remotesensing, doi:10.3390/rs14153832_

Round 1

Reviewer 1 Report

Wide-area InSAR deformation monitoring has become a research and application hotspot. This study proposes a wide-area InSAR variable scale deformation detection strategy, the WAVA-InSAR strategy, which combines the Stacking technology for fast ground deformation rate calculation and the advanced TS-InSAR technology for obtaining fine time-series deformation. Moreover, the authors propose a wide-area deformation products organization structure to generate the variable-scale deformation products. The proposed methods are verified in the Turpan-Hami basin, China. The strategy proposed in this manuscript is innovative in application with an adequate amount of work. So, I think the present manuscript is suitable for publication in “Remote Sensing”.

The following problems exist in the manuscript and need to be improved.

1、The title of Section 2.1 is inappropriate. Please confirm whether there is a problem with typesetting;

2、L39: monitor the deformation over wide area;

3、The term “time series deformation” in the main text should be replaced with “time series of deformation” or “deformation time series”, e.g., L47, L51, L90;

4、L82: “has” less information;

5、L151: In “order” to improve …;

6、L178: I would like to use multi-look number instead of multi-look ratio, as the ratio may be equal, such as 10:2 and 5:1;

7、L270: “the main image” is should be “the master image”;

8、L280: spatial baseline? Do you mean perpendicular baseline?

9、Section 4.2.3 compare the consistency of results from different frames. Although this can be used to evaluate the reliability of the InSAR results. But this may not be called “accuracy assessment”, perhaps “reliability assessment” would be more appropriate;

10、Following the logic of the strategy that stacking to obtain the wide-area deformation velocity and ROI detection, TS-InSAR to refine the deformation in the ROI, what does the ALOS-1 data play in the processing?

Reviewer 2 Report

The manuscript by Wang et al. is well-planned work. I think it is worth publishing in Remote Sensing. I have only some minor comments. Before publishing the manuscript, please consider them.

1.     The authors should not use abbreviations which is not in common, such as WSA and SFM.

2.     The authors should add the major mountains and fault’s names, which they mentioned in the main text, in Figure 2 because I cannot understand which part of the figure was mentioned.

3.     I guess that Figure 3 shows the LOS (Line of Sight) velocities observed by ascending orbits of Sentinel-1. However, the authors only mentioned it as the deformation rate map. I recommend that the authors should add more detailed information to the figure. For example, what do the negative and positive values (warm and cold colors) show in the figure?

4.     The areas of deformation funnels are shown in Table 2, but I cannot understand where each deformation funnel is located. I recommend that the authors add the numbers of deformation funnels in Figure 3.

5.     In Figure 4, the author should add explanations of what the negative and positive values show in each sub-figure.

6.     Although the vertical axis of Figures 5 and 6 are subsidence, it should be LOS displacement. If the authors calculated vertical deformation from the LOS displacements, assuming all of the LOS displacement is vertical displacement, the authors should mention it in the captions.

7.     The authors should add the detail of optical images which was used in Figures 8, 9, and 11. For example, please add the satellite, sensors, and capture date of images.

8.     In Line 515, the title of section 5.3 does not match the contents.

Author Response

This manuscript is a resubmission of an earlier submission. The following is a list of the peer review reports and author responses from that submission.